# Determination of 14 Isoflavone Isomers in Natto by UPLC-ESI-MS/MS and Antioxidation and Antiglycation Profiles

**DOI:** 10.3390/foods11152229

**Published:** 2022-07-26

**Authors:** Aoli Xiang, Jingyi Wang, Bijun Xie, Kai Hu, Mengting Chen, Zhida Sun

**Affiliations:** 1Department of Food Science and Technology, Huazhong Agricultural University, Wuhan 430070, China; nasax@webmail.hzau.edu.cn (A.X.); bijunxie@mail.hzau.edu.cn (B.X.); hzau2015hukai@foxmail.com (K.H.); chenmengting@webmail.hzau.edu.cn (M.C.); 2College of Bioengineering and Food, Hubei University of Technology, Wuhan 430068, China; jywang@hbut.edu.cn; 3Laboratory of Food Technology, Department of Microbial and Molecular Systems (M2S), KU Leuven, 3001 Leuven, Belgium

**Keywords:** natto, isoflavone, isomer, antioxidant, anti-glycation

## Abstract

Natto is a famous traditional fermented food, but the influence of the fermentation process on the content and composition of soybean isoflavones and nutritional value is still unclear. In the present study, the variation in soybean isoflavones during fermentation by *Bacillus subtilis* natto was revealed by UPLC-ESI-MS/MS (Ultra high performance liquid chromatography-electron spray ionization-mass spectrometry) analysis. After 24 h of fermentation, the total isoflavone content in natto increased by 1.62 times compared with fresh soybean, and the content of aglycones was 3.07 times that of raw beans. More importantly, among 14 isoflavone isomers identified in natto, the isomers of daidzin, genistin, and succinyl genistin were detected for the first time, which might be due to the result of isomerase and succinylase and other corresponding enzymes’ action in *Bacillus subtilis*. In addition, natto isoflavones performed great antioxidant activity than its monomer components (glycosides daidzin and genistin, aglycones genistein and daidzein), except for genistein. Moreover, natto isoflavone and its aglycones (especially genistein) performed great inhibitory activity against AGEs (Advanced Glycation End Products) in three in vitro models. The mechanism test showed that genistein could form adducts (UPLC-Q-TOF-ESI-MS/MS analysis) with methylglyoxal. These findings demonstrated that soybean fermented with *Bacillus subtilis* natto had a significant influence on the isoflavone profiles and its bioactivity.

## 1. Introduction

Natto is a traditional food fermented from soybean by *Bacillus subtilis* [1,2], and it has been widely and stably consumed in Asia for centuries. Since the discovery that soy consumption is associated with a reduced risk of breast cancer in women due to bioactive ingredients such as isoflavones (genistein, daidzein, and glycitein), fermented natto is thought to be more nutritious and have more health benefits than raw soybeans, although its acceptability in western countries is limited due to its distinct aroma and flavor.

Over the last four decades, more than 10,000 global scientific papers and reviews have appeared on isoflavones, and these publications have identified the structure and several benefits of isoflavones in the diet [3]. As one of the most important effective ingredients in soybean, isoflavones are bioactive compounds of non-steroidal and phenolic nature with the mother nucleus of 3-benzopyranone, with mildly estrogenic properties and often referred to as phytoestrogen [4,5,6,7,8]. There is a total of 12 kinds of soybean isoflavones naturally existing in soybean, which can be divided into three categories, namely daidzin groups, genistin groups, and glycitin groups. Each type exists in four forms: free aglycones, glucoside type, acetyl glucoside type, and malonyl glucoside type (Figure 1), and it must be noted here that malonyl glucosides almost are hydrolyzed and converted into highly active aglycones in the fermentation process. Spontaneously, the strategies to enhance the content of total isoflavones in soybean-based food have received greater attention in both laboratory and industry settings, and the treatment of soybean by microorganisms (fermentation) has been proved to be one of the safe and effective tools to achieve this goal [9].

According to recent studies, the fermentation procedure could change the isoflavone profiles and greatly boost the biosynthesis and bioavailability of soybean isoflavones, especially the content of aglycones with higher bioactivity and digestibility [10,11]. The greatest increases were found when using *Pseudomonas fluorescens* (1.28-fold) and Stenotrophomonas maltophilia (1.32-fold), whereas *Aspergillus sojae* allowed the greatest increase in aglycon isoflavones (8.74-fold), with increases of 12.15-, 11.23-, and 1.17-fold for daidzein, genistein, and glycitein, respectively. More interestingly, we also found that the isoflavone glycosides (daidzin and genistin) in natto fermented were not only modified but also isomerized, forming a pair of glycoside isomers at C4′ and C7 positions of B ring and A ring, which were previously only found in human metabolism, but have not been reported in natto. These isomers showed significant biological activity and might provide a better utilization value of soybean-based food. As a result, this bio-transform exerts beneficial effects on menopausal symptoms, metabolic-related diseases, and cancers [12]. For example, in a previous report published by Toda [13], soybeans fermented with *Bacillus subtilis* (natto) not only biosynthesized several new isomers of isoflavone (6″-*O*-acyl isoflavone glycosides and 6″-*O*-succinylated isoflavone glycosides), the fermented food also possessed great preventive effects on bone loss in ovariectomized rats fed a calcium-deficient diet. However, compared with the widely studied nattokinase and menaquinone-7, soybean isoflavones in natto have received less attention. To the best of our knowledge, the structure and modification of isoflavones in natto and its anti-organic free radical and glycosylation inhibition features are relatively few reports.

Advanced Glycation End Products (AGEs) are stable end products of a non-enzymatic reaction involving a reducing sugar carbonyl and a free amino group of an amino acid or protein, lipids, nucleic acid, and other macromolecular components [14], and as toxic compounds, the accumulation of AGEs in body fluids contributes to toxic pathogenesis, such as Alzheimer’s disease, atherosclerosis, cardiovascular diseases, cancer, Lipid metabolism disorder, cataract, retinopathy, inflammation, and oxidative stress [15,16]. In recent years, the development of natural AGEs inhibitors has become a hot spot. As natural antioxidant agents, flavonoids, such as quercetin, genistein, and gallic acid, are efficient glycoxidation inhibitors that have been extensively investigated [17,18]. As previously reported, soybean isoflavones represented by genistein mainly inhibit the production of AGEs by capturing dicarbonyl compounds [19]. The inhibitory activity of soybean isoflavones on glycation is affected by chemical structure and is proportional to its antioxidant activity [20]. Therefore, the effects of isomerization on the isoflavone profile during soybean-based food fermentation and changes in anti-organic free radical and glycosylation inhibition activity alteration worth to be investigating.

In order to solve the above issues, the purpose of the present paper was to finely analyze the composition of soybean isoflavone in natto by *Bacillus subtilis* and identify its chemical structure through UPLC-ESI-MS/MS(Ultra high performance liquid chromatography-electron spray ionization-mass spectrometr) analysis, especially analyzing the formation of isoflavone isomer and its succinylated derivatives for the first time in detail. In addition to describing the nutritional composition of natto, antioxidant, and anti-glycation activities of purified soybean isoflavones in vitro were emphatically studied. Results obtained through the present paper will provide an experimental basis and new ideas for the utilization and health significance of natto.

## 2. Materials and Methods

### 2.1. Chemicals and Materials

Daidzein, glycitein, genistein, daidzin, glycitin, and genistin were purchased from Yuanye Biotechnology Co., Ltd. (Shanghai, China) with a purity ≥98%. Methanol (HPLC grade) was purchased from Thermo Fisher Scientific (Shanghai, China). All other chemicals and reagents were analytical grade, and reverse-osmosis Milli-Q water was used.

The fresh soybeans were purchased from the local supermarket. *Bacillus subtilis* natto ATCC 65301 used for fermentation was purchased from American Type Culture Collection (ATCC) and maintained in 20% glycerol at −80 °C. Its cultural conditions and activation methods are applied according to the report of Li [21].

### 2.2. The Fermentation of Soybean and Isoflavone Preparation

Before fermentation, 200 g of fresh soybean was soaked in 600 mL water for 6 h, and the damaged and non-foaming beans were removed. Then, the selected soybeans were sterilized at 115 °C for 30 min. After natural cooling, the activated *B. subtilis* strain solution (10 mL, 1.24 × 10^12^) was sprayed on the 100 g soybean surface in the sterilized fermentation box, and the soybean was rotated with a spoon to ensure complete and even inoculation. The fermentation process was maintained at 37 °C for 24 h and then placed in a 4 °C environment for 12 h (post-ripening stage) [22].

Then, freeze-dried natto was grounded into powder for degreasing by Soxhlet extraction. Place the freeze-dried natto powder in a filter paper bag and extract it for 12 h with petroleum ether at a boiling range of 30 to 60 °C; then, the fat-free natto freeze-dried powder was obtained. Through single-factor orthogonal experiments, the optimal extraction process parameters of natto isoflavone were determined as follows: 1.0 g defatted natto powder was mixed with 35 mL methanol solution (80%, *v*/*v*) and stirred at 50 °C for 150 min, and the extraction process was repeated three times, and then the supernatant was collected and combined. After removing methanol, crude soybean isoflavones were obtained through freeze-dried treatment. In addition to a small number of soybean isoflavones required in natto, huge content of water, protein, water-soluble polysaccharide, ash, and so on needed to be removed by AB-8 macroporous resin. As a polystyrene weak polar adsorption resin, AB-8 may bind strongly hydrophobic soybean isoflavones. After dissolving the crude soybean isoflavone extract in water, the sample was loaded into the column at a ratio of 1:5, adsorbed by AB-8 macroporous resin, and washed with distilled water at a ratio of 5 times the column volume to remove soluble proteins and polysaccharides, and then the Coomassie brilliant blue method and the molish reaction, respectively, determined whether the proteins and polysaccharides were completely removed. Finally, elute the soybean isoflavones with 95% ethanol after eluting the resin column with 20% ethanol and twice the column capacity. After purification, the darkest and clearest liquid column was gathered as the finished good.

Therefore, the traditional chemical method and AB-8 macroporous resin were utilized according to Xi [23] to achieve the goal of purification, and the purity of soybean isoflavones (μg of β-glucoside equivalent/mL) was determined with the reference of genistin and daidzin according to the protocol provided by Yanaka [24]. The content of soybean isoflavones was then determined by the normalization method.

Daidzin and genistin accounted for 97% of the total content of soybean isoflavones, and the ratio of genistin to daidzin was 3:1. An ultraviolet-visible (UV-vis) spectrophotometer (UV-1800, Shimadzu, Kyoto, Japan) was utilized to detect the absorbance at 260 nm. Two calibration curves for the daidzin and genistin standard (at concentrations of 0.002, 0.004, 0.006, 0.008, 0.010 mg/mL) was prepared. The linear regression equation were y_daidzin_ = 0.0928x_1_ + 0.0399 (R^2^ = 0.9993) and y_genistin_ = 0.118x_2_ − 0.005 (R^2^ = 0.9990).
Isoflavone concentration (μg of β-glucoside equivalent/mL) = 1/4 y_daidzin_ + 3/4 y_genistin_.

The total isoflavone content in natto was calculated as the sum of the contents of individual isoflavone molecular species.

### 2.3. Characterization of Soybean Isoflavone from Natto

#### 2.3.1. FT-IR (Fourier Transform Infrared Spectoscopy) Analysis

The freeze-dried soybean isoflavone powder was mixed with KBr in a 1:50 ratio (*w*/*w*) and ground. Then, 100 mg of ground powder was pressed into tablets for FT-IR analysis using an iS50R FT-IR spectrometer (Thermo Scientific, Waltham, MA, USA). The sample scanning data were collected across a full waveband (4000–400 cm^−1^) at a resolution of 4 cm^−1^.

#### 2.3.2. HPLC and UV Analysis

Waters e2695 HPLC (Waters Alliance, NE, USA)system coupled with a photodiode array (PDA) detector (Waters 2998 DAD) (Waters Alliance, NE, USA) set at a wavelength of 260 nm, and a ZORABX SB-C18 (Agilent Technologies Inc., CA, USA) reversed-phase column (250 × 4.6 mm, 5 μm) was used for soybean isoflavone qualitative analysis at room temperature. The mobile phase consisted of acetonitrile containing 0.025% acetic acid (A) and 0.025% acetic acid–water (B), using the following linear gradients: 0–70 min: 5–25% A, and then held at 25% for 3 min; 73–80 min: 25–5% A, and then held at 5% for 3 min. Mobile phase B changes according to the change in mobile phase A and keeps A + B = 100%. The flow rate was 1.0 mL/min, and the injection volume was 10 μL.

In order to acquire the UV absorption scanning spectra and the greatest absorption peak, daidzein, daidzin, glycitin, genistin, and genistein were dissolved in methanol, and glycitein was dissolved in dimethyl sulfoxide (DMSO). The UV absorption spectra of six types of common soybean isoflavones were conducted by purchased standard, and the remainder of the UV absorption spectra were extracted from the HPLC spectra. UV scanning was performed after the sample solution was dissolved in methanol, and the scanning curves were recorded in the spectrum of 200–400 nm with an ultraviolet-visible (UV-vis) spectrophotometer (UV-1800, Shimadzu, Kyoto Japan).

### 2.4. UPLC-ESI-MS/MS Analysis of Soybean Isoflavone from Natto

Thermo scientific ultimate 3000 UPLC and Q exactive electrostatic field orbital hydrazine mass spectrometry offered the isoflavone MS data. As described previously, UPLC was used to separate the sample. MS condition: atmospheric pressure chemical ionization (ESI) ion source in both positive and negative modes, spray voltage of 3200 V, capillary temperature of 300 °C, sheath gas of 40.00 Arb, Aux gas of 8.00 Arb, Max spray current of 100.00 μA, and Probe heater temperature of 280 °C. Full MS/dd–MS^2^ mode was used to capture the fragment pattern.

### 2.5. Antioxidant Capacity of Isoflavones from Natto

The antioxidant capabilities of isoflavones extract from natto were determined and compared with four kinds of common isoflavone standards, including daidzin, daidzein, genistin, and genistein.

#### 2.5.1. DPPH Radical Scavenging Capacity

According to a previous report by Chen [25], soybean isoflavone samples with varying concentrations (1.0–5.0 mg/mL) were treated with a DPPH free radical dissolved in ethanol (0.05 mol/L) at a 1:2 volume ratio. The absorbance value at 570 nm was measured after 30 min of the light avoidance reaction. As a blank group and control group, anhydrous ethanol was used instead of sample solution and DPPH free radical. Daidzin, daidzein, genistin and genistein were used as positive controls. The DPPH free radical scavenging rate was calculated using the equation below:Scavenging rate (%) = (1 − (A_i_ − A_j_)/A_0_) × 100
where A_i_, A_j_, and A_0_ represent the absorbance of the soybean isoflavone sample, control group, and blank group, respectively.

#### 2.5.2. ABTS Radical Scavenging Capacity

ABTS radical scavenging activity was assessed following the method of Re [26]. ABTS and potassium persulfate solutions of 7 mmol/L and 2.45 mmol/L, respectively, were prepared using distilled water. The two solutions were mixed in the same volume and reacted in the dark for 12 h. The reaction solution was prepared after diluting 50 times with anhydrous ethanol. Soybean isoflavone solution with various concentrations (1.0–5.0 mg/mL) was mixed with 2.4 mL ABTS working solution for 10 min in the darkness, and then measured the absorbance at 734 nm. The control group utilized distilled water instead of ABTS working solution, and the blank group used methanol instead of soybean isoflavones. Daidzin, daidzein, genistin, and genistein was used as positive control.
Scavenging rate (%) = (1 − (A_i_ − A_j_)/A_0_) × 100
where A_i_, A_j_, and A_0_ represent the absorbance of the soybean isoflavone sample, control group, and blank group, respectively.

#### 2.5.3. O_2_^−^ Radical Scavenging Capacity

The determination of superoxide anion radical is carried out according to the instructions of the kit named Inhibition and produce superoxide anion assay kit (A052-1-1) from Nanjing Jiancheng Bioengineering Institute (Nanjing, China). The experimental results are expressed by the change value of superoxide anion radical inhibited by 1 mg vitamin C equivalent as an activity unit.

#### 2.5.4. Ferric Reducing Antioxidant Power (FRAP) Capacity

The FRAP of soybean isoflavone was determined using the method of Benzie and Strain with small modifications [27]. An amount of 25 mL 0.3 mol/L acetate buffer (pH 3.6) was mixed with 2.5 mL TPTZ solution (2.4.6-Tris(2-pyridyl)-s-triazine) (0.01 mol/L) and ferric chloride solution to produce the FRAP reagent (0.02 mol/L). The FRAP reagent was isometrically mixed with soybean isoflavones methanol solution (0.2–1.0 mg/mL), and the absorbance value at 593 nm was recorded after 10 min of light, avoiding a reaction. Instead of FRAP working solution, the control group utilized distilled water, and the blank group used sample solvent (methanol) instead of soybean isoflavones.

### 2.6. In Vitro Inhibition of Advanced Glycation End Products by Soybean Isoflavones from Natto

In order to determine the inhibition rate of soybean isoflavones on AGEs, three systems were established: The physiological environment was simulated by the α-lactose–lysine system (α-lactose-Lys system) and by using fructose–bovine serum albumin system (fructose-BSA system) to imitate a food environment. In addition, the bovine serum albumin–methylglyoxal system (BSA-MGO system) was also established to explore the inhibition rate of soybean isoflavones on AGEs.

#### 2.6.1. Fructose-BSA System

In order to simulate the inhibitory effect of soybean isoflavones on AGEs in the physiological system, the fructose-BSA system was selected. A fructose-BSA model was used to monitor the process of BSA protein glycation based on the study by Shen [20] with slight modification. Briefly, 1.5 mol/L fructose PBS solution (0.5 M, pH = 7.4) was mixed with sample solutions (0.2–1.0 mg/mL) and then placed in a water bath at 37 °C for 2 h. Afterward, 30 mg/mL BSA-PBS solution was added and received another 24 h incubation at 37 °C. The fluorescence intensity was measured at a wavelength of 370 nm for excitation and 450 nm for emission. Daidzin, daidzein, genistin, and genistein were utilized as a positive control. The AGEs inhibition rate was calculated by the following formula:inhibition rate (%) = (1 − F_1_/F_0_) × 100 
where F_0_ and F_1_ represent the fluorescence intensity of the blank group and isoflavone sample group, respectively.

##### 2.6.2. α-lactose-Lys System

α-lactose-Lys system was selected to simulate the inhibitory effect of soybean isoflavones (0.2–1.0 mg/mL) on AGEs in the food system. An amount of 0.1 mol/L α-lactose solution and 0.1 mol/L lysine solution with PBS (0.5 mol/L, PH 7.4) were prepared, respectively. Lactose was mixed with sample solutions and then placed in a water bath at 37 °C for 2 h, followed by adding Lys’s solution and placing in a water bath at 37 °C for 24 h. The fluorescence intensity was measured at a wavelength of 370 nm for excitation and 450 nm for emission. Daidzein was utilized as a positive control. The AGEs inhibition rate was calculated by the following formula:inhibition rate (%) = (1 − F_1_/F_0_) × 100
where F_0_ and F_1_ represent the fluorescence intensity of the blank group and isoflavone sample group, respectively.

#### 2.6.3. BSA-MGO System

In order to assess the middle stage of protein glycation, the BSA-MGO model was used [28]. An amount of 60 mM MGO solution and 30 mg/mL BSA solution were prepared with PBS (0.5 mol/L, PH 7.4), respectively. MGO was mixed with sample solutions (0.2–1.0 mg/mL) and then placed in a water bath at 37 °C for 2 h. Afterward, 30 mg/mL BSA-PBS solution was added and received another 24 h incubation at 37 °C. The fluorescence intensity was measured at a wavelength of 370 nm for excitation and 450 nm for emission. Daidzin, daidzein, genistin, and genistein were utilized as a positive control. The AGEs inhibition rate was calculated by the following formula:inhibition rate (%) = (1 − F_1_/F_0_) × 100
where F_0_ and F_1_ represent the fluorescence intensity of the blank group and isoflavone sample group, respectively.

### 2.7. Mechanism of Genistein Inhibiting AGEs in BSA-MGO System

Through the above experiments, isoflavones exerted the highest inhibition effect in the BSA-MGO system, as well as genistein showed the best preventive capacity among all isoflavone quercetins. Therefore, the MGO-BSA system was selected to explore the inhibitory effect and mechanism on AGEs by genistein.

#### 2.7.1. Kinetic Study of the Trapping of MGO by Genistein

MGO (60 mM) was incubated with 200 μg/mL genistein in a phosphate buffer solution (0.05 mol/L PH 7.4) at 37 °C for 0, 10, 30, 60, 120, 240, 480, 720 and 1440 min. HPLC was used to determine the amounts of MGO [17]. Briefly, the waters e2695 HPLC system coupled with a photodiode array (PDA) detector (Waters 2998 DAD) set at a wavelength of 260 nm, and a ZORABX SB-C18 reversed-phase column (250 × 4.6 mm, 5 μm) was used with a flow rate of 0.5 mL/min. The mobile phase consisted of 10% methanol aqueous solution containing 0.2% acetic acid (A) and methanol containing 0.2% acetic acid (B), using the following linear gradients: 0–1.5 min: 5–20% B, and 1.5–2.5 min: 20–35% B, finally 2.5–4.5 min: 35–100% B, then held at 100% for 1 min.

#### 2.7.2. Kinetic Study of the Inhibitory Effects on the Formation of AGEs by Genistein

Since genistein was determined as the most effective natto isoflavone in trapping MGO, we further selected genistein to evaluate its kinetics of AGEs inhibition. BSA (30 mg/mL) was incubated with MGO (60 mM) in the presence or absence of genistein (200 μg/mL) in a phosphate buffer solution (0.05 mol/L PH 7.4). The reaction mixture was collected at different time points (240, 480, 720, 960, 1200, and 1440 min). Fluorescence at an excitation/emission wavelength of 370/450 nm, which is characteristic of AGEs, was used to measure AGEs levels.

#### 2.7.3. UPLC-Q-TO-ESI-MS/MS Analysis of Genistein-MGO Adducts

UPLC/MS analysis was carried out with a Waters Vion IMS Q-TOF Spectra System incorporated with an electrospray ionization (ESI) interface. A ZORABX SB-C18 reversed-phase column (250 × 4.6 mm, 5 μm) was used for separation at a flow rate of 0.37 mL/min [29]. The column was eluted with 90% solvent A (acetonitrile with 0.1% formic acid) and linear increases in solvent B (water with 0.1% formic acid) to 30% from 0 to 4.08 min, maintaining 30% from 4.08 to 5.44 min, and then with 90% A from 5.44 to 6.26 min, maintaining 90% from 6.26 to 6.53 min. The LC eluent was introduced into the ESI interface. The ESI ion source was tuned to negative ion polarity mode, with the voltage on the ESI interface kept at 2 kV. The sheath gas was utilized at a flow rate of 45 arb units, while the auxiliary gas was used at a flow rate of 5 arb units. By using collision-induced dissociation (CID) with a relative collision energy setting of 35%, tandem mass spectrometry (MS/MS) was used to gather structural information on genistein and the primary MGO adducts.

### 2.8. Statistical Analysis

The data are expressed as mean and standard deviation. Analysis of variance (One-Way ANOVA) and Duncan’s multiple range tests was used to determine the difference between the results (*p* < 0.05).

## 3. Results

### 3.1. Nutrient Components Analysis of Natto and Isoflavones Preparation

As shown in Table 1, the content of protein, carbohydrate, and fat was enhanced after fermentation. More importantly, the content of total isoflavones in crude extract significantly (*p* < 0.05) increased by 61.53% in natto compared with the fresh soybean, and aglycone content also increased significantly. Changes in isoflavone content before and after fermentation were determined by HPLC mentioned in the following Section 3.2.1. As is shown in Figure 2, among all isoflavones detected in natto, β-glucoside accounted for 72.72%, acetyl isoflavones accounted for 7.40%, and aglycones and succinyl isoflavones produced by fermentation accounted for 4.39% and 15.49%, respectively. Compared with that before fermentation, the content of soybean isoflavone aglycones and succinyl glucosides was significantly increased, whereas the content of glucosides significantly decreased. During the process, 6″-*O*-malonyl-glycitin disappears, meanwhile new 6”-*O*-succinyldaidzin and 6″-*O*-succinylgenistin were formed.

Moreover, through single-factor and orthogonal tests (Appendix A), the total content of soybean isoflavones was determined to be 2.88 ± 0.44 mg/mL, and the final purity of soybean isoflavones from natto increased from 1.42% to 76.76% after the purification process.

### 3.2. Spectral Characterization of Soybean Isoflavone in Natto

The spectral characteristics of soybean isoflavones in natto fermentation by *Bacillus* were characterized by UV, HPLC, IR, and UPLC-ESI-MS/MS. A total of 14 compounds were identified, including three kinds of soybean isoflavone products, named aglycone, glycoside, and acylation. Among them, succinyl isoflavone glycosides were unique acyl compounds produced during natto fermentation.

#### 3.2.1. HPLC and UV Analysis

UV characteristic absorption peaks and HPLC analysis results of aglycone and glycoside standard of soybean isoflavones and natto extract were shown in Appendix A (See Section 3.2.3 for product analysis of peak 4, peak 7, peak 8, peak 9, and peak 11 except for the six known standards). Because soy isoflavones have a strong conjugated system of hydroxyl and aromatic rings in their structure, therefore soy isoflavones showed characteristic UV absorption peaks at 260 nm and 320 nm (Figure 3). The UV spectrums of natto isoflavones extract were consistent with the characteristic absorption spectrum of isoflavone standards (see Figure 4). UV spectrophotometry is often used to measure the level of soy isoflavones in samples.

#### 3.2.2. FT-IR Analysis

FT-IR analysis of soybean isoflavones extracted from natto is shown in Figure 5. The strong absorption peaks at 3454 cm^−1^ and 3238 cm^−1^ were attributed to the stretching vibration of hydroxyl groups with different degrees of association. The C-H stretching vibration occurred at 2919 cm^−1^, and the C-H stretching vibration for aromatic compounds was at 3079 cm^−1^. The skeletal vibration absorption peak of the benzene ring appeared at 1621 cm^−1^, 1579 cm^−1^, 1519 cm^−1^, and 1450 cm^−1^, respectively. The strong absorption peak at 1659 cm^−1^ indicated stretching vibration of C=O in the 3-benzopyranone of soybean isoflavone head. Bands at 1271 cm^−1^ attributed to the in-plane bending vibration absorption peak of =C-O-C on γ-pyranone. The absorption peak at around 1183 cm^−1^ was related to the in-plane bending vibration of C-O, and the in-plane bending vibration of C-H happens at 1087 cm^−1^. As shown by the absorption peaks of the tested sample, the molecule had the characteristic functional groups of soybean isoflavone. The above attribution is consistent with the values calculated by the reported density generic function theory [30].

#### 3.2.3. UPLC-ESI-MS/MS Analysis

Table 2 shows the results of UPLC-ESI-MS^2^ identification of soybean isoflavones in natto. Fourteen isoflavone isomers were observed in positive ion mode, including three kinds of soybean isoflavone compounds, namely, aglycones, glycosides, and acylates. Additionally, the sequence of each aglycone, glucosides, acetyl glucosides, and succinyl glucosides detected on the C18 column was consistent with the conclusions of previous studies [31]. Among them, the isomers of daidzin, genistin, and succinyl genistin were detected for the first time in *Bacillus subtilis* fermentation natto. These results indicated that isomerization and succinylation of soybean isoflavones have occurred during the natto fermentation process. Detailed analysis results are as follows:

MS/MS fragmentation of peaks corresponding to compounds **1**, **2** resulted in the production of a common fragment at *m*/*z* 255 (417–162), which is the parent ion losing a glucose molecule. Their molecular weights are 417.11783 and 417.11795, respectively, and the errors of the theoretical values are 1.78 and 1.79 ppm. These compounds displayed different retention times on the HPLC column; thus, they indicated the isomers of daidzin. There are two conjugation sites on daidzein at C-4′ and the C-7 site (Appendix A). It was established that the C-7-glucoside of isoflavones is retained longer on reverse-phased sorbents as compared to their C-4′-isomers. Based on ultraviolet (UV) chromatograms and peak intensities observed in the MS/MS chromatogram (Appendix A), peak 2 was identified as daidzein-7-glucoside, and peak 1 was identified as the daidzein-4′-glucoside [32]. The molecular ions *m*/*z* 447.12851, 433.11310, and the corresponding fragments (285.07574 and 271.06021) correspond to peaks 3 and 4, respectively, and the error with the theoretical value and UV spectral characteristics, indicating that compounds **3** and **4** are glycitein and genistein-4′-glucoside, respectively. Peak 5 is an isomer of genistin [32]. The molecular ions, *m*/*z* 517.13416 to peak 6 is a characteristic compound 6″-*O*-succinyldaidzin produced by *Bacillus subtilis* fermented natto, which has the characteristic fragment of daidzein 255.06514, fragmentation ions (255.06514, 285.07571 and 271.06012), and *m*/*z* (459.12869, 489.13922, and 475.12372) of peaks corresponding to compounds **7**, **8**, and **13** attributable to acetylation conjugates (6″-*O*-acetyldaidzin, 6″-*O*-acetylglycitin and 6″-*O*-acetylgenistin), and peaks 9, 11 were tentatively identified as two succinylation conjugates (6″-*O*-succinylgenistin) isomers, respectively. Additionally, *m*/*z* (255.06509, 285.07587, and 271.06006) of peaks corresponding to compounds **10**, **12**, and **14** attributable to aglycones (daidzein, glycitein, and genistein). The sequence of each aglycone, glucosides, acetyl glucosides, and succinyl glucosides detected on the C18 column was consistent with the conclusions of previous studies. These results indicate that the isomerizations and succinylation of soybean isoflavones mainly occur in daidzein and genistein compounds; thus, isomerase and succinylase can exist in *Bacillus subtilis*, which need to be further identified. The isomer structural formula is shown in Appendix A.

Heat treatment during fermentation can reduce the ratio of malonyl-isoflavones in soybean but enhance the ratio of aglycone and glycoside. Meanwhile, heat and microorganism treatment degrade and break the binding between isoflavones and macromolecular substances in soybean, releasing the free isoflavones. Interestingly, succinyl soybean isoflavones, which are unique in natto, were found in the sample after fermentation, and this alteration of the isoflavone profile in natto might bring a better activity.

### 3.3. Antioxidant Capacity of Soybean Isoflavones from Natto

In soybean isoflavones, The B ring is located at the 3-position, which are structural isomers of flavonoids, so this has antioxidant activity. Differences in molecular structures and electron distributions affect the antioxidant capabilities of phenolic compounds with electron- and hydrogen-donating abilities [33].

On the basis of previous experiments, we found that the aglycone activity of soybean isoflavones was significantly higher than the corresponding glycosides, and the activities of genistein and daidzein in the four aglycones were higher. Therefore, in this study, five experimental samples of glycosides, aglycones, and isoflavone extracts from natto were selected, and their antioxidant activities were evaluated by four different antioxidant methods. The results are shown in Figure 6.

In the DPPH system, the inhibition effect of aglycone on DPPH was better, and the effect of genistein was the best. However, their effective inhibitory concentrations were all above 1.0 mg/mL. This is probably attributable to the glucose moieties linked to the isoflavone backbone having a higher steric barrier than the isoflavone aglycones [34], which prevents them from binding to organic anion radical DPPH.

However, in the organic cation free radical system, natto isoflavone extract and soybean isoflavone monomer showed an excellent inhibitory effect. Among them, the extract has the best effect. This is due to the charge effect of the acylation products generated after fermentation. In general, the order of ABTS inhibition ability of experimental samples is as follows: isoflavone sample exacted from natto > genistein > daidzein > genistin > daidzin. It can be seen that after aglycone glycosylation, the anti-free radical ability of aglycone decreases due to the reduction in the number of hydroxyl groups on the A ring and the increase in polarity and steric hindrance. In addition, the higher the number of hydroxyl groups on the benzene ring, the stronger its antioxidant capacity is, so the activity of dye lignin with three hydroxyl groups is higher than that of daidzein with two hydroxyl groups.

In the ABTS radical scavenging assay, each soybean isoflavone showed a stronger scavenging ability than DPPH radicals. Among these, the antioxidant ability of genistein was similar to that of the extracted samples. At lower concentrations (0.05 mg/mL~0.1 mg/mL), the extracted sample had a higher ability to scavenge ABTS radicals than genistein. In general, the antioxidant activity is as follows: isoflavone sample exacted from natto > genistein > daidzein > genistin > daidzin. The trend of decreasing antioxidant capacity with the increasing molecular weight of soy isoflavones may be attributed to the steric hindrance as indicated in the DPPH assay, and its antioxidant activity is inversely proportional to the size of molecular weight.

For the FRAP system, the antioxidant capacity for soybean isoflavones decreased as follows: genistein > daidzein > isoflavone sample exacted from natto > genistin > daidzin, indicating that antioxidant capacity depends on the functional moieties bound to 3-benzopyrone parent nucleus. Compared with the previous FRAP experiment, the antioxidant capacity of aglycones was consistent [35]. Additionally, in the present experiment, genistin showed a stronger antioxidant capacity than daidzin, which is consistent with the findings of previous studies [10].

As we all know, isoflavones’ ability to scavenge O_2_^−^ free radicals are linked to their structure. The crucial component in isoflavones that provides them antioxidant properties is the hydroxyl group on the B ring and A ring. All kinds of items could scavenge O_2_^−^ free radicals structurally [36]. However, the ability to scavenge superoxide anion radicals varies due to differences in steric hindrance and solubility produced by the varied molecular weights of each sample. This was proved by our experimental results. Isoflavone glycosides have a significantly lower O_2_^−^ inhibition effect than aglycones due to steric hindrance caused by conjugation (Figure 6D). The antioxidant capacity of test samples for the superoxide anion radical scavenging experiment is genistein > isoflavone sample exacted from natto > daidzein > genistin > daidzin. The effect of glycoside soybean isoflavones on scavenging superoxide anion free radicals is poor (less than 50 U/L). The experimental results are consistent with those of the DPPH experiment. This is closely related to their charge effect.

### 3.4. Anti-Glycation Activity of Soybean Isoflavones from Natto

The above results determined that soybean isoflavones are powerful natural antioxidants. Previous studies showed that AGEs formation is negatively correlated with antioxidant activity against free radicals. Therefore, we reported the AGE’s inhibitory properties of genistein, daidzein, genistin, daidzin, and natto isoflavone extracts in three models in vitro of the AGEs inhibition effect. The results are shown in Figure 7.

The fructose-BSA model is commonly used to evaluate the inhibitory effect of natural products on AGEs in vitro. As is shown in Figure 7A, in this system, isoflavone standard and extract showed a strong inhibitory effect on the formation of AGEs, and there was a dose–response relationship. Among them, genistein has the best effect. When its concentration was only 200 μg/mL, the inhibition rate reached 50%, and when the concentration increased to 1.0 mg/mL, the inhibition rate was 75.63% (Figure 7A). The activity of natto isoflavone extract was second only to genistein. This is related to the highly active succinyl daidzin and succinyl genistin products after fermentation.

The order of their inhibitory activity was genistein > sample > daidzein > genistin > daidzin. Anti-glycation activity appeared to be more dependent on the chemical structure of the aglycone, in agreement with the conclusion of Wang et al. [19]. The effect is consistent with the experimental result of the antioxidant test. ROS can interact with the side chains of protein or amino acid residues to increase the content of carbonyl proteins and mercaptans. The mechanism lies in the effective inhibition of ROS by isoflavones.

The lactose–lysine model was set up to mimic the generation of AGEs in the food system. Free amino group residues, especially lysine, arginine, and cysteine (the N-terminal amino group can α-dicarbonyl compounds form adducts), and glycosylation produce irreversible protein modification, thus reducing its nutritional value. As a reducing sugar, lactose can be widely ingested in the daily diet and easily oxidized to form α-dicarbonyl compounds. Therefore, a lactose–lysine model could be used to assess the antiglycation capability of soybean isoflavones in preventing α-dicarbonyl from binding to lysine. In the current research, on the contrary, the effect of glycosides is more significant than that of aglycones (Figure 7B). Aglycones only show a weak inhibitory effect, which is contrary to the experimental results of the first two models. The inhibitory effect of flavonoids extracted from natto is significantly better than that of glycosides and aglycones, which may be the result of the joint action of a variety of flavonoids. As the extract sample concentration increased to 0.6 mg/mL, the inhibition rate exceeded 50%. The conclusion of this model is consistent with that of Genova [37].

Methylglyoxal (MGO) has high reactivity among the many reactive dicarbonyl compounds and AGEs precursors. It is believed to contribute significantly to intracellular AGE formation. MGO could readily react with arginine, lysine, and cysteine residues of proteins, modify proteins, and damage DNA. Additionally, the concentration of MGO in diabetes mellitus was found to be increased 2- to 6-fold [38].

As is shown in Figure 7C, the MGO-BSA model is of great significance for evaluating the level of AGE and the development of diabetes. In the BSA-MGO system, all isoflavones samples showed a strong inhibition effect on the formation of AGE, with the best activity of genistein, and its maximum inhibitory concentration reached 95.31% at 1.0 mg/mL [29]. Mechanism studies showed that the inhibition effect of genistein for AGE was due to the genistein trapping of MGO by the formation of mono- and di-adducts between genistein and MGO, which prevented the reaction of MGO with protein.

Then, the capture effect of genistein on MGO and the inhibition effect on AGEs were analyzed by the kinetic study, and the mechanism of genistein inhibiting AGEs was analyzed by UPLC-Q-TOF-ESI-MS/MS analysis [17,39,40].

### 3.5. Mechanism of Genistein Inhibiting AGEs in BSA-MGO System

#### 3.5.1. Kinetic Study of the Inhibitory Effects on the Formation of AGEs by Genistein

In order to provide more detailed inhibitory parameters, we further evaluate the efficiency that genistein trapped MGO (Figure 8A). Within 240 min, more than 80% of MGO was trapped, with trapping efficiency reaching 97.1% after 1440 min. Meanwhile, genistein can significantly inhibit the generation of AGEs in the BSA-MGO system (Figure 8B). The inhibition rate of AGEs can reach more than 85% when the reaction time reaches 1440 min.

#### 3.5.2. UPLC-Q-TOF-ESI-MS/MS Analysis

The reaction mixtures of MGO and genistein (1:1 ratio) were incubated at 37 °C for 24 h. After incubation for 24 h, two new major peaks (RT 3.49 and 3.79 min) were observed. Figure 9A depicts a retention time of 3.79 min with the molecular ion *m*/*z* 341 [M − H]^−^. The peak was 72 mass units higher than genistein (*m*/*z* 269 [M − H]^−^), indicating the mono-MGO adduct of genistein. The peak at 3.49 min in Figure 9B had the fragment ion *m*/*z* 413 [M − H]^−^, which had 72 mass units more than the mono-MGO adduct of genistein, indicating that it is the di-MGO adduct of genistein. This result is in line with prior findings [17,29].

## 4. Conclusions

Through a systematic study of natto isoflavones, current results showed that the microorganism process was a feasible technical method for improving the nutritional value and bioactivity of soybeans. Due to the fermentation, the content of total isoflavones in natto increased by 1.62 times that in soybean, which further enhances the health value of natto. Meanwhile, UPLC-ESI-MS/MS analysis was applied to analyze the bio-transform of natto isoflavone, and 14 isoflavone isomers were identified. Among them, daidzin, genistin, and succinyl genistin were detected for the first time, and it indicated the existence of related enzymes and bio-reactions. Based on these structure change analyses, the antioxidant and anti-glycation activity of natto isoflavones and their main monomers were tested for the first time, and both natto isoflavone and genistein owed relative higher bioactivities. Therefore, our research laid a theoretical foundation for the utilization of active isoflavones and the development of fermented soybean-based food in the food industry.

## Figures and Tables

**Figure 1 foods-11-02229-f001:**
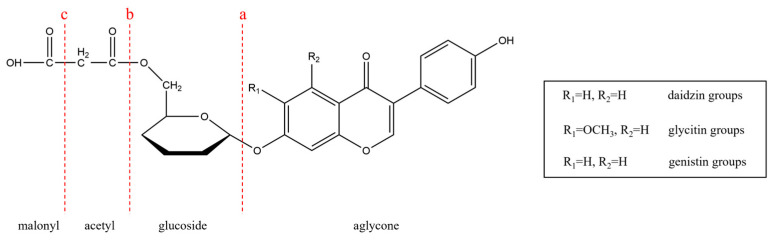
Chemical structures of 12 natural isoflavones.

**Figure 2 foods-11-02229-f002:**
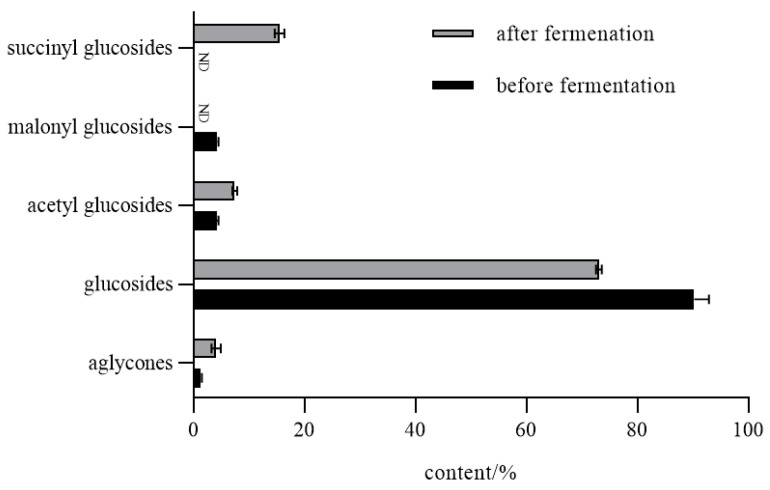
Changes in flavone content before and after fermentation. ND means not detected.

**Figure 3 foods-11-02229-f003:**
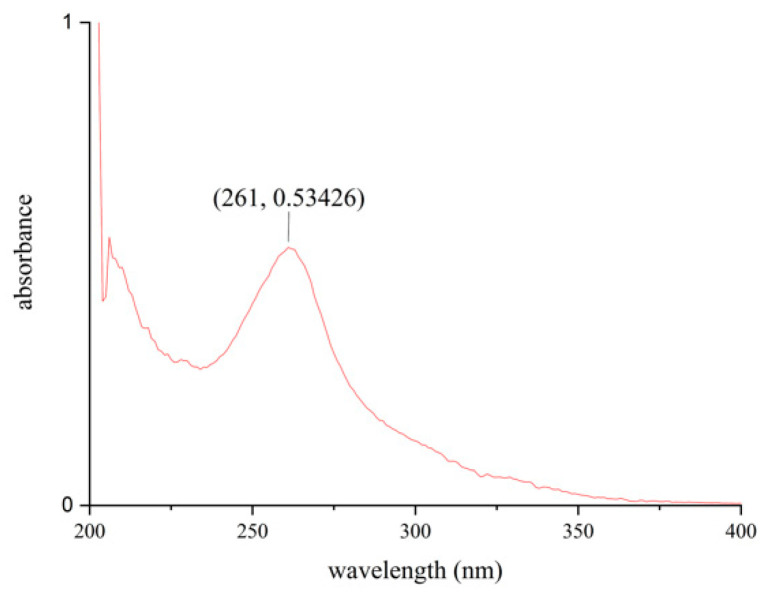
UV scanning spectra of isoflavone extracts from natto.

**Figure 4 foods-11-02229-f004:**
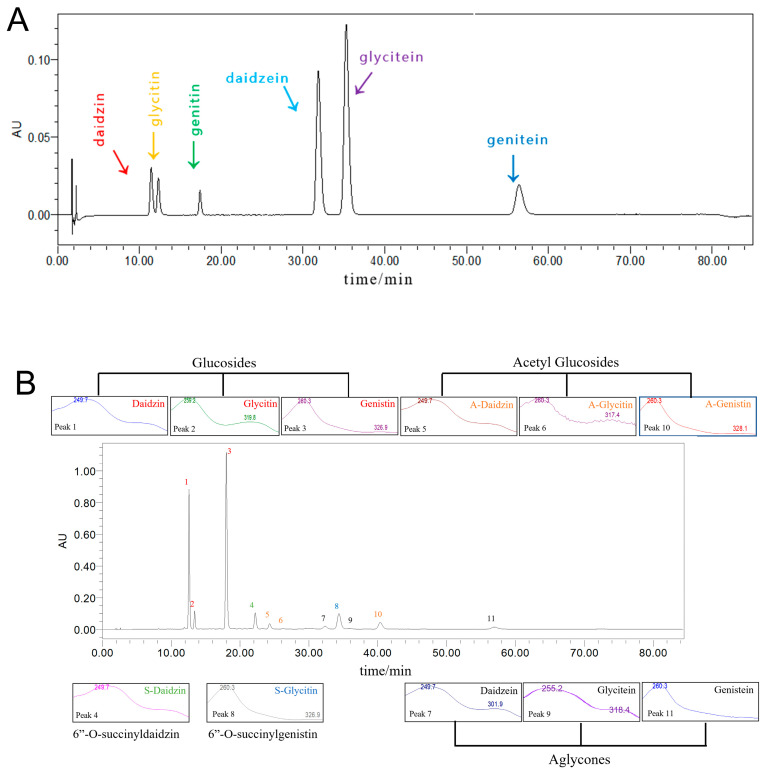
HPLC diagram of mixed soybean isoflavone standards (**A**) and HPLC chromatograms of the sample purified and standards of soy isoflavones by AB-8 macroporous resin (**B**), as recorded at 254 nm. The numbers of the peaks are as follows: 1. Daidzin; 2. Glycitin; 3. Genistin; 4. 6″-*O*-succinyldaidzin; 5. 6″-*O*-acetyldaidzin; 6. 6″-*O*-acetylglycitin; 7. Daidzein; 8. 6″-*O*-succinylgenistin; 9. Glycitein; 10. 6″-*O*-acetylgenitin; 11. Genistein.

**Figure 5 foods-11-02229-f005:**
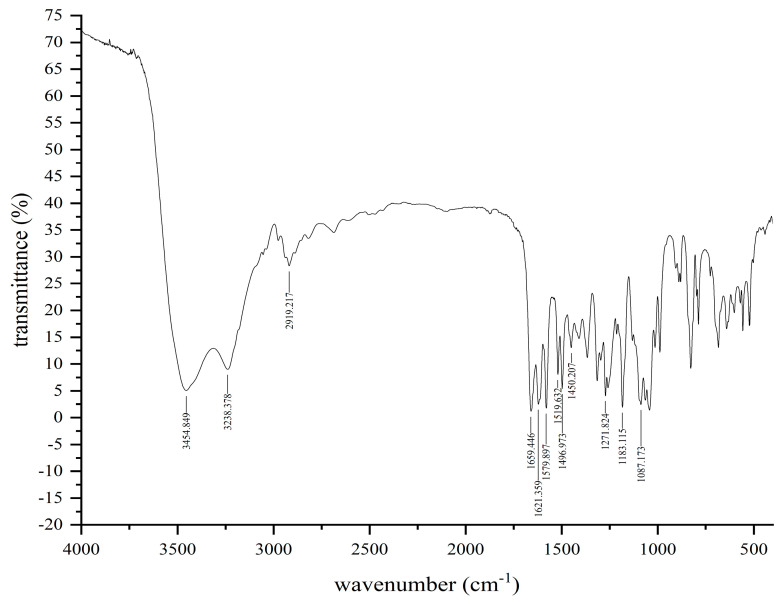
FT−IR scanning of extract sample.

**Figure 6 foods-11-02229-f006:**
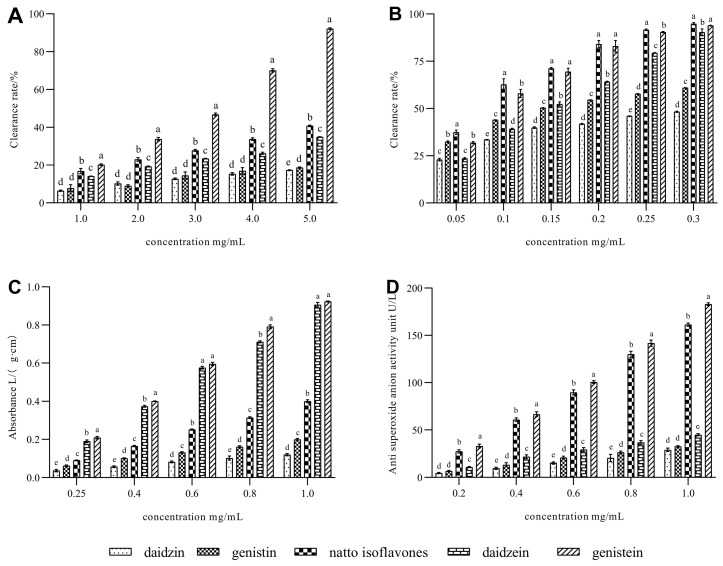
Antioxidant capability of various isoflavone standards and the extract isoflavone from natto. (**A**) DPPH radical scavenging rate; (**B**) ABTS radical scavenging rate; (**C**) Ferric Reducing Antioxidant power capacity; (**D**) O_2_^−^ radical scavenging capacity. Data are presented as the means ± SD of three replications. Different letters represent significant differences (*p* < 0.05).

**Figure 7 foods-11-02229-f007:**
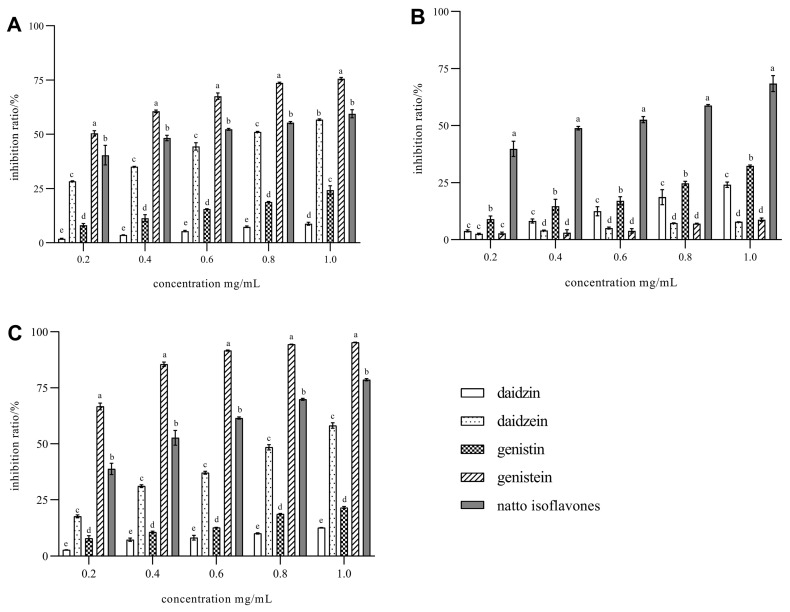
Determination of AGEs inhibition rate in vitro model. (**A**) fructose-BSA model; (**B**) lactose–lysine system; (**C**) MGO-BSA system. Data are presented as the means ± SD of three replications. Different letters represent significant differences (*p* < 0.05).

**Figure 8 foods-11-02229-f008:**
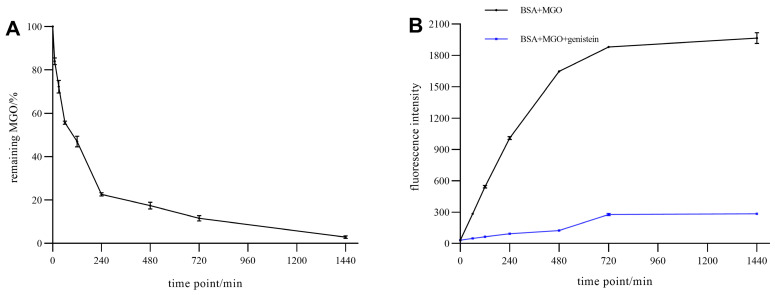
(**A**) Trapping of MGO by genistein under physiological environment (PH 7.4, 37 °C) and (**B**) inhibitory effect of the formation of AGEs by genistein in the BSA-MGO system.

**Figure 9 foods-11-02229-f009:**
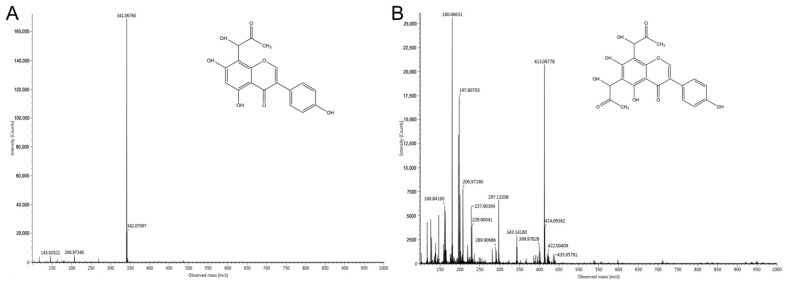
Tandem MS spectra of mono-MGO and di-MGO adducts of genistein. (**A**) MS spectrum of mono-MGO adduct; (**B**) MS spectrum of di-MGO adduct.

**Table 1 foods-11-02229-t001:** Analysis of natto nutrition component changes (dry basis/%) and antioxidant capacity.

	Protein (%)	Fat (%)	Carbohydrate (%)	Ash (%)	Total Isoflavones (mg/kg)	ABTS ^1^	Superoxide Anion ^2^
Before fermentation	40.80 ± 0.13 ^b^	15.75 ± 0.21 ^b^	7.43 ± 0.32 ^b^	4.46 ± 0.39 ^a^	1487.23 ± 34.22 ^b^	17.15 ± 0.24% ^b^	n.d. ^3^
After fermentation	43.25 ± 0.25 ^a^	20.85 ± 0.07 ^a^	27.33 ± 1.65 ^a^	4.40 ± 0.4 ^a^	2402.65 ± 55.87 ^a^	44.27 ± 1.27% ^a^	25 ^4^

^1^ 2,2′-Azino-bis (3-ethylbenzothiazoline-6-sulfonic acid) radical scavenging assay; ^2^ super oxygen-anion free radical scavenging ability; ^3^ not detected; ^4^ Data are expressed as mean ± standard deviations (*n* = 3). Different letters in same row represents significant differences (*p* < 0.05).

**Table 2 foods-11-02229-t002:** UPLC-ESI-MS/MS analysis of isoflavone sample extracted from natto.

Peaks	TR ^a^ (min)	Compounds	Formulas	Formula Weight	Precursor Ions [M + H]^+^	Exact Mass [M + H]^+^	Mass Error ^b^	λ_max_ (nm)	Fragment Ions (*m*/*z*)
1	11.13	daidzein-4′-glucoside	C_21_H_20_O_9_	416	417.11783	417.11037	1.78	250.0, 305.0	417.11673/255.06509
2	12.72	daidzein-7-glucoside	C_21_H_20_O_9_	416	417.11795	417.11037	1.79	250.0, 304.0	417.11658/255.06525
3	13.54	glycitin	C_22_H_22_O_10_	446	447.12851	447.12130	1.61	259.2, 319.8	447.12766/285.07574
4	15.36	genistein-4′-glucoside	C_21_H_20_O_10_	432	433.11310	433.10565	1.72	262.0, 327.5	433.11438/271.06021
5	18.42	genistein-7-glucoside	C_21_H_20_O_10_	432	433.11298	433.10565	1.69	262.0, 326.0	433.11438/271.06021
6	23.10	6″-*O*-succinyldaidzin	C_25_H_24_O_12_	516	517.13416	517.12678	1.43	249.7	517.13409/255.06514
7	25.33	6″-*O*-acetyldaidzin	C_23_H_22_O_10_	458	459.12869	459.12130	1.61	249.7	459.12933/255.06514
8	27.53	6″-*O*-acetylglycitin	C_24_H_24_O_11_	488	489.13922	489.13186	1.50	260.3, 317.4	489.13754/285.07571
9	31.51	6″-*O*-succinyl-4′-genistin	C_25_H_24_O_13_	532	533.12921	533.12169	1.41	262.0, 326.0	533.12982/271.06030
10	33.86	daidzein	C_15_H_10_O_4_	254	255.06509	255.05791	2.82	249.7, 301.9	255.06511/-
11	36.35	6″-*O*-succinyl-7-genistin	C_25_H_24_O_13_	532	533.12933	533.12169	1.43	262.0, 325.0	533.12946/271.06021
12	37.55	glycitein	C_16_H_12_O_5_	284	285.07587	285.06847	2.59	255.2, 318.4	285.07581/-
13	42.88	6″-*O*-acetylgenistin	C_23_H_22_O_11_	474	475.12372	475.11621	1.58	260.3, 328.1	475.12369/271.06012
14	59.93	genistein	C_15_H_10_O_5_	270	271.06006	271.05282	2.67	260.3	271.060188/-

^a^ Retention time; ^b^ Difference between observed mass and exact mass error (<5 ppm).

## Data Availability

The data presented in this study are available on request from the corresponding author. The data are not publicly available due to privacy.

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
