# Peer review of "Determination of 14 Isoflavone Isomers in Natto by UPLC-ESI-MS/MS and Antioxidation and Antiglycation Profiles"

_foods, 2022, doi:10.3390/foods11152229_

Round 1

Reviewer 1 Report

1-    Lines 58-59: add the full name of the microrganisms (P. fluorescens, S. maltophilia, and A. sojae).

2-    Line 112: make B. subtilis italic

3-    Line 113: is this correct (10 mL, 1.24×1012) or change it to (10 mL, 1.24×1012).

4-    Line 125: please add full detailed method you used to remove protein, fat, water-soluble polysaccharide from the extract instead of this short sentence (the traditional chemical method and AB-8 macroporous resin were utilized ac-125 cording to Xi [23])

5-    Line 126: please add full detailed method you used to determine soybean isoflavones (μg of β- glucoside equivalent/mL)

6-    Please add full detailed method (to help the others to reprocess this method) for the following sections; 2.5.1. DPPH radical scavenging capacity, 2.5.2. ABTS radical scavenging capacity, 2.5.3. O2·- radical scavenging capacity (add the name and number of the kit), 2.5.4. Ferric Reducing Antioxidant Power (FRAP) capacity, 2.6.1. Fructose-BSA system, 2.6.2. α-lactose-Lys system, 2.6.3. BSA-MGO system.

7-    Line 306: please add the full methods from which you calculated the values in this figure (Figure 2. Changes in flavone content before and after fermentation).

In the results of HPLC analysis (Figure 4.) you used only 6 standards as in Figure 4A, but in Figure 4B you characterized extra 5 peaks, so please add the methods you used to identify these extra peaks?? 

Reviewer 2 Report

In the study entitled "Determination of 14 Isoflavone Isomers in Natto by UPLC-ESI-MS / MS and Antioxidation and Antiglycation Profiles" which studies natto isoflavones, the current results showed that the microorganism process was a feasible technical method to improve the nutritional value and soybean bioactivity. .

Due to fermentation, the total isoflavone content of natto increased 1.62 times that of soy, which further increases the health value of natto.

Meanwhile, the UPLC-ESI-MS / MS analysis was applied to analyze the biotransformation of natto isoflavone and 14 isoflavones were identified. Among them, daidzine, genistin and succinyl genistin were detected for the first time and indicated the existence of enzymes and related bio-reactions. Based on these structural modification analyses, the antioxidant and anti-glycation activity of natto isoflavones and their main monomers were tested for the first time, and both natto isoflavones and genistein were given relatively higher bioactivities.

The manuscript is interesting, but needs a little improvement. The purpose of this study is not described in the abstract. I recommend adding purpose in the abstract.

The introduction also provides sufficient information on the subject.

The Materials and Methods section is detailed corespunzator. The results are well organized and detailed. They are easy to follow and clearly presented. Statistical analysis seems to be appropriate.

And the discussion section is detailed, clear and appropriate. The conclusion is short, concise and to the point.
